

# Sedentary patterns and health outcomes in the oldest-old: a latent profile analysis

Elien Lebuf[1,2], Delfien Van Dyck[1], Laura Van de Velde[1], Melanie Beeckman[1], Jelle Van Cauwenberg[2] and Sofie Compernolle[1,3]

[1] Department of Movement and Sports Sciences, Ghent University, Ghent, Belgium
[2] Department of Public Health and Primary Care, Ghent University, Ghent, Belgium
[3] Research Foundation Flanders, Brussels, Belgium

## ABSTRACT

**Background.** Sedentary behavior is most prevalent among those aged 80 years and above, referred to as the oldest-old. Current literature emphasizes the significance of sedentary behavior patterns, but further evidence is required to understand how these patterns relate to specific health outcomes and to identify at-risk profiles for tailored interventions in the oldest-old. Therefore, the aim of this study was to identify profiles of adults aged 80+ years based on their sedentary patterns and health outcomes, and to examine associations between profiles and socio-demographics.

**Methods.** A cross-sectional study was conducted in Flanders (Belgium) from February 2021 to December 2022 recruiting 90 older adults (80+) through convenience sampling, employing word of mouth, social media and local service centers. Latent profile analysis identified device-based sedentary patterns and assessed their associations with physical and cognitive functioning, mental health-related quality of life (QoL), and social isolation. Associations of these profiles with socio-demographic factors were analyzed.

**Results.** Three distinct profiles were identified: (1) the 'cognitively and physically frail' profile, (2) the 'healthy' profile and (3) the 'lower mental health-related QoL' profile. Those in the 'cognitively and physically frail' profile exhibited the least favorable sedentary pattern, and had a higher likelihood of residing in a nursing home. No significant differences were found for the other socio-demographic variables, being age, sex, educational degree and family situation.

**Conclusions.** Three distinct profiles in the oldest-old population, based on cognitive and physical functioning, mental health-related QoL, and sedentary behavior patterns, were identified. Lower physical and cognitive functioning was associated with unhealthy sedentary patterns. Further research with larger samples is crucial to uncover potential links between socio-demographics and at-risk subgroups, enhancing our understanding of sedentary behavior and geriatric health outcomes in this population.

Corresponding author
Sofie Compernolle,
sofie.compernolle@ugent.be

# INTRODUCTION

Sedentary behavior (SB) is defined as "any waking behavior characterized by an energy expenditure of ≤1.5 metabolic equivalents (METs), while in a sitting, reclining or lying posture" (*Sedentary Behaviour Research, 2012*). Growing evidence identifies SB as an important risk factor for detrimental health-related conditions, including type 2 diabetes,

metabolic syndrome, cardiovascular disease, cancer, and increased risk of mortality in both adult and older adult populations (*de Rezende et al., 2014*; *Ekelund et al., 2019*; *Katzmarzyk et al., 2019*; *Wilmot et al., 2012*). Older adults (65 years and older) are the most sedentary age group, spending on average 9.4 hours/day sedentary (*Harvey, Chastin & Skelton, 2013*; *Harvey, Chastin & Skelton, 2015*). Moreover, a recent meta-analysis showed the importance of distinguishing between these age groups. This study revealed that individuals in the oldest-old age group (80+) spend an average of one to two additional hours/day in sedentary activities compared to a combined group of participants aged 60 years or older, which includes those aged 80 and above (*Webster et al., 2021*), and underlines the importance of studying SB specifically in the oldest-old.

Recent studies emphasize the significance of 'sedentary accumulation patterns' over the mere total sedentary time (ST) in relation to various health-related outcomes. ST is defined as the time spent in sedentary activities, while accumulation patterns encompass how sedentary activities accumulate or gather over a specific period of time. Measures of these accumulation patterns may involve the duration, frequency, and distribution of ST across the day or week (*Boerema et al., 2020*). In addition to the negative correlation between overall ST and health outcomes, extended sedentary bouts, characterized as uninterrupted periods of ST (*Tremblay et al., 2017*), have been linked to diminished physical function in older adults (*Gennuso et al., 2016*; *Han et al., 2022*). Similarly, older women with a higher mean sedentary bout duration have increased risk of falling (*Rosenberg et al., 2021*) and show higher odds of diabetes (*Bellettiere et al., 2019*). Another study suggests that for every additional hourly break in ST in older women, the odds for abdominal obesity decreased by 7% (*Júdice et al., 2015*). Furthermore, a study in community-dwelling older men found that those who spent significantly more time in prolonged sedentary bouts were more likely to be depressed, obese and to suffer from chronic disease(s) (*Jefferis et al., 2015*).

Previous studies examining patterns of SB and their associations with health outcomes have mainly focused on the general population of older adults aged 65 years and older, whereas very few studies focused specifically on those aged ≥80 years. However, SB patterns tend to change within age groups from 65 to ≥90 years, as older age seems to be associated with more ST, fewer sit-to-stand transition and longer usual sedentary bout (*Dohrn et al., 2020*). Consequently, there is a critical need for further investigation into how SB patterns among the oldest-old relate to health outcomes. This underrepresentation is not only present in research studies but is also reflected in the guidelines on physical activity and SB set by the World Health Organization (WHO).

Moreover, existing literature studying the association of SB with health outcomes in older adults predominantly focuses on physical health aspects, particularly chronic diseases. It is imperative, however, to extend our studies to include mental, cognitive and social dimensions of health given their inherent impact on independence and quality-of-life (*Dogra et al., 2022*). To achieve a comprehensive understanding, it is recommended to adopt a broader, multidisciplinary approach. This approach involves integrating insights from various disciplines, such as medicine, psychology, and sociology, to gain a holistic perspective. This multidisciplinary perspective creates the possibility to effectively identify profiles of older adults at risk and develop interventions aimed at improving sedentary

patterns and overall well-being. Considering that studies have emphasized the significance of socio-demographic factors in influencing the interplay between sedentary patterns and health outcomes (*Chastin et al., 2015*; *Copeland et al., 2017*), determining the link between socio-demographics and these at-risk profiles of SB patterns and health outcomes in the oldest-old can provide valuable insights.

In conclusion, ST is highest in the oldest-old (80+) and current literature points towards the importance of patterns in SB. Still, more evidence is needed to unravel how certain SB accumulation patterns cluster with specific health outcomes to identify at-risk profiles for future SB interventions in the oldest-old. Therefore, the aim of this study is to identify person-centered profiles, using a Latent Profile Analysis (LPA), based on accumulation patterns of SB and the following health outcomes: physical, cognitive and psychosocial functioning, in the oldest-old (≥80 years). After identifying the different profiles, possible predictors of profile membership based on socio-demographic variables will be investigated to gain further insight into the characteristics of older adults in the most vulnerable profiles.

## METHODS

### Study design

A cross-sectional study, called 'Healthy 80+', was performed between February 2021 and December 2022 in Flanders, Belgium. The study was approved by the ethical committee of the Ghent University Hospital (BC-08641), and all participants provided a written informed consent. This paper follows the STROBE guidelines for reporting observational studies (*von Elm et al., 2014*). Portions of this text were previously published as part of a preprint (https://www.researchsquare.com/article/rs-3352579/v1).

### Participants and procedure

Older adults aged 80 years or older were recruited for this study through convenience sampling. Recruitment was done through word of mouth, social media and local service centers. Local service centers offer information and recreational activities within neighborhoods to facilitate self-reliance, especially among people for whom their care or medical requirements have recently changed. Participants had to be Dutch speaking, able to stand independently with or without an assistive device and could not have been diagnosed with cognitive disorders to be eligible for the study. Participants who were willing to engage in the study were contacted by phone to arrange the dates for the two home visits. The first home visit consisted of explaining the purpose and procedures of the study, signing the informed consent, completing a questionnaire during a structured interview and executing the Short Physical Performance Battery. At the end of the first home visit, an activPAL accelerometer was attached to the participants thigh to measure SB for seven consecutive days. Participants were also asked to keep a sleep diary. On day eight, the second home visit was planned to recollect the activPAL accelerometer and the sleep diary, and to perform a cognitive functioning test.

## Measures

### Sedentary behavior

*Accelerometer data processing:* An activPAL accelerometer (Physical Activity Technologies, Glasgow, Scotland) was attached on the midline of participants' right anterior thigh for seven consecutive days to objectively measure ST. The activPAL is an accelerometer that is known as the gold standard for objectively measuring ST since it can distinguish between sitting/lying and standing (*Hart, Ainsworth & Tudor-Locke, 2011*). Data were summarized in epochs of 15 s and were downloaded and processed using manufacturer proprietary software (activPAL 173 Professional v8.11.6.70). A day was considered invalid if there was limited postural variation (*i.e.,* ≥95% of wear time in one activity) or fewer than 10 hours of valid waking wear time (*Morris et al., 2019*; *Winkler et al., 2016*). Only participants with four days of valid activPAL data were included in the analysis (*Heesch et al., 2018*). The 'Processing PAL' (*Winkler et al., 2016*) application was used to manually double-check the algorithms to exclude sleep from the analysis. Therefore, participants were asked to report the times they went to bed and when they got up in the morning in their sleep diary. If there was an incongruence between self-reported and algorithm-derived sleep time, adaptations were made using the 'Processing PAL' application based on the self-reported data (*Winkler et al., 2016*). Likewise, the created heatmaps from the 'Processing PAL' application were used to manually check the algorithm for potentially misclassified invalid days, since the algorithm seems to be less reliable in older adult populations (*Winkler et al., 2016*).

*Total sedentary time and sedentary accumulation patterns:* Total ST (minutes/day) was extracted from the accelerometer data to maximize comparability with other studies. For identifying SB patterns many measures exist, but consensus about the best indicators is still lacking (*Boerema et al., 2020*). Alpha (unitless) is one of the most robust measures and is very sensitive to change (*Chastin et al., 2015*). Alpha is defined as "the cumulative distribution of bout lengths" and represents the frequency distribution of SB bout duration, which follows the power-law probability distribution (*Chastin & Granat, 2010*; *Chastin et al., 2015*). Therefore it acts as a measure that captures the diversity of bout lengths during a day (*Boerema et al., 2020*). Lower values indicate a SB accumulation pattern with more prolonged bouts, whereas higher values represent a more fragmented SB pattern. One disadvantage of using alpha, is that it can be difficult to interpret. Therefore, a second sedentary accumulation pattern measure, usual bout duration (UBD), was calculated. UBD is the median value of the cumulative sedentary bout duration distribution, and is a universal measure to report bout lengths in relation to total ST (*Boerema et al., 2020*). All three measures: total ST, alpha and UBD, were calculated using Python. The script can be found at OSF (https://osf.io/g2cn7/).

### Physical functioning

During the first home visit, the Short Physical Performance Battery (SPPB) was performed by trained researchers. This test battery consists of three physical performance tests (balance, walking, sit-to-stand) and is a well-established tool to evaluate functional capability (*Guralnik et al., 1994*). For each test a subscore ranging from 0 to 4 was created. A summary score for physical functioning was obtained by summing the three subscores,

resulting in a score ranging from 0 to 12, with higher scores indicating better physical functioning. A detailed summary of the scoring system can be found elsewhere (*De Fátima Ribeiro Silva et al., 2021*).

### Cognitive functioning

The Cambridge Neuropsychological Test Automated Battery (CANTAB) was performed during the second home visit. This battery focuses on three cognitive domains: (1) working memory and planning, (2) attention and (3) visuospatial memory.

The CANTAB is the most widely published battery for cognitive function, is largely independent of verbal instruction and is a relatively cheap and accessible method to assess cognitive functioning compared to face-to-face assessments (*Lenehan et al., 2016*; *Smith et al., 2013*). The CANTAB was performed using an iPad. To make sure participants were familiar with the device and understood all instructions, a familiarization exercise was performed first (motor screening task, MOT). After that, a subset of five tests was performed: reaction time (RTI, attention and psychomotor speed), paired associates learning (PAL, memory), spatial working memory (SWM, executive functions), delayed matching to sampling (DMS, memory) and rapid visual information processing (RVP, attention and psychomotor speed).

First, $Z$-scores for all outcomes (*e.g.*, reaction time, error percentage) were calculated (one for RTI, two for PAL, SWM and DMS, one for RVP; $N = 8$). Second, negatively scored outcomes were reversed so that all were positive scores with a higher score indicating better cognitive functioning. Third, for the cognitive tests with multiple outcomes, composite scores were calculated by averaging the Z-scores of all respective outcomes. To calculate overall cognitive functioning of the participants, an average score of all composite scores was calculated (*Gheysen, Herman & Van Dyck, 2019*).

### Mental health-related quality of life

Mental health-related quality of life (QoL) was derived from the RAND-36 questionnaire, assessed during the first home visit. First, scores for the following subscales for mental health were calculated: vitality, social functioning, emotional role functioning and mental health (*Ware, 1994*). Second, the sum of these scores was divided by 4, which resulted in the unweighted RAND-36 Mental Composite Summary (MCS). This RAND-36 MCS is a simple, validated way of interpreting the mental health aspect of the RAND-36 (*Andersen et al., 2022*).

### Feelings of social isolation

The Patient-Reported Outcome Measurement Information System (PROMIS) short form for social isolation was assessed. Four items concerning feelings of social isolation were asked, with five answer possibilities ranging from 'never' to 'always'. PROMIS item banks and validated short-forms are created to enable researchers to get insight into participants' symptoms, functioning and health-related quality of life in an efficient, flexible and precise way (*Cella et al., 2010*). Scoring was done according to PROMIS scoring guidelines: a sum-score of all items was calculated to represent 'feelings of social isolation', which could only be calculated with complete data. From this raw sum-score, a T-score with mean of

50 and standard deviation of 10 was calculated using the score conversion table (*PROMIS, 2021*).

### Socio-demographic, BMI and co-morbidities

The following variables were obtained *via* a questionnaire during the first home visit: age (years); sex (male or female); weight (kg) and height (cm), from which BMI was calculated; marital status (single, in a relationship, cohabiting/married); living situation (home, service flat, nursing home); and education (no degree, primary education, vocational secondary education, technical secondary education, general secondary education, higher education and university; which was recoded into 'no higher education' or 'higher education'). Also comorbidities (NSHAP Comorbidity Index, measuring burden of chronic diseases and conditions) were assessed by asking "has a medical doctor told you that you have (had) [condition]?" for a list of 15 conditions, with a range from 0 to 21; scoring details can be found in *Vasilopoulos et al. (2014)*.

## Statistical methods

All analyses were executed using RStudio (R version 3.4.1), with the script and anonymized data available on the OSF page (https://osf.io/g2cn7/). Descriptive statistics covered socio-demographic data, BMI and comorbidities, SB measures, and health outcomes.

Latent profile analysis (LPA), using the 'tidyLPA' R package (*Rosenberg et al., 2018*), was performed to identify latent subgroups among older adults. This person-centered approach explores varied patterns of variables and outcomes (*Muthen & Muthen, 2000*). The LPA procedure (*Bauer, 2022*) involved the following steps:

First, theoretically-based indicators were selected, *i.e.,* mean ST/day, alpha and UBD (SB accumulation patterns), global cognitive functioning (CANTAB), mental health-related quality-of-life (unweighted RAND-36 MCS), feelings of isolation (PROMIS) and physical functioning (SPPB). Second, inter-correlation analyses were performed, identifying UBD as more correlated to the other indicators compared to mean ST/day and alpha, leading to their exclusion from the model. Third, negatively scored indicators (*i.e.,* PROMIS and USB) were scored positively. Fourth, possible models were compared using the following fit indices: Bayesian Information Criterium (BIC), sample-size adjusted BIC (SABIC) and Akaike Information Criterion (AIC) (lower values indicate a better model fit to the data), the Bootstrap Likelihood Ratio Test (BLRT) and entropy values (indicates the overall ability of a model to return well-separated profiles, higher value means better fit with one being perfect classification). Selecting the optimal model involves assessing both the aforementioned fit indices and the theoretical framework relevant to the discipline. This requires careful consideration to ensure that the identified solution aligns with possible real-world profiles. Fifth, the target solution was interpreted by generating standardized scores for the included variables, examining class-specific means, probability profiles and class sizes. Sixth, post-LPA, one-way ANOVA or non-parametric Kruskal–Wallis analyses were conducted to assess differences in SB and health outcomes among profiles. *Post-hoc* tests to identify pairwise differences between profiles included Tuckey's HSD and Dunn's test respectively.

To assess profile differences in socio-demographics, one-way ANOVA and Chi$^2$ analysis compared age, sex, education, living situation, and family situation. Fisher's exact test was used for small subgroup observations (*i.e.*, less than five) (*Bower, 2003*; *McCrum-Gardner, 2008*). Socio-demographic variables included: age (continuous), sex (1 = male, 2 = female), education (1 = no higher education; 2 = higher education), living situation (1 = living at home or service flat; 2 = nursing home), family situation (1 = living alone; 2 = living together with partner). Significance was set at $p \leq 0.05$.

## RESULTS

### Characteristics of study population

A total number of 112 participants were included in the study. However, after exclusion due to incomplete CANTAB ($N = 8$) and activPAL data ($N = 14$), analyses were performed on 90 participants. Errors during the activPAL setup, specifically placing the device back into the charger after completion of the initialization, led to missing activPAL data in 14 subjects at the beginning of the study.

Socio-demographic, SB and health-related characteristics can be found in Table 1. The study sample had a mean (SD) age of 84.9 ± 4.0 years, with a maximum of 95.0 years. About half of the participants were women (53%) and about one third of the population had a high education level (30%). The majority of participants lived independently at home (83%) with their partner (55%). About one quarter were underweight (25%), 44% had a healthy weight, and 32% were overweight or obese. The mean (SD) sitting time of the participants was 9.6 ± 1.9 hours/day, with a mean (SD) UBD of 44.4 ± 21.2 min.

### Identified profiles in the oldest-old

Considering the principles of parsimony ('if two options are plausible, the simpler one is preferred'), the interpretability of profiles, and the evaluation of multiple fit indices across different models (*Bauer, 2022*), the results of the LPA indicated that a three profile model provided the best solution for this study. See Table 2 for details.

Figure 1 shows group profiles based on the five indicators: UBD, physical and cognitive functioning, mental health-related QoL and feelings of social isolation. Profiles one, two and three included 12% ($N = 11$), 76% ($N = 68$) and 12% ($N = 11$) of the sample, respectively. Those proportions are in line with the rule of thumb that each profile should include >1.0% of the total sample size (*Lubke & Neale, 2006*). Additionally, the 3-profile solution, considering the differences in ST and health outcomes, could be interpreted as follows: Profile one, labeled as 'Cognitively and Physically Frail' and comprising 12% ($N = 11$) of the sample, exhibited longer bouts of SB (with 10 min longer UBD compared to the second profile), significantly lower physical and cognitive functioning, and a lower mental health-related QoL. Profile two, labeled as 'Healthy' and representing 76% ($N = 68$) of the sample, demonstrated a healthier SB pattern, significantly better physical and cognitive functioning, and higher mental health-related QoL. Profile three, labeled as 'Lower Mental Health-Related QoL' and accounting for 12% ($N = 11$) of the sample, displayed similar

**Table 1  Sample characteristics.**

| Demographics | Total sample ($N = 90$) | Missing (N) |
|---|---|---|
| Age, mean (SD) | 84.9 (4.0) | |
| Women (%) | 53.3 | |
| Marital status | | 1 |
| Single (%) | 42.7 | |
| In a relationship, not living together (%) | 2.2 | |
| Cohabiting/married (%) | 55.1 | |
| Participants without children (%) | 10.0 | |
| Highly educated (%) | 30.0 | |
| Living situation | | |
| Living at home (%) | 83.3 | |
| Living at assisted-living apartment (%) | 7.8 | |
| Living at nursing home (%) | 8.9 | |
| BMI, mean (SD) | 26.1 (3.9) | 1 |
| Underweight (%) | 24.7 | |
| Healthy weight (%) | 43.8 | |
| Overweight (%) | 28.1 | |
| Obese (%) | 3.4 | |
| Comorbidity index[1], mean (SD); median (IQR) | 2.5 (1.6); 2.0 (2.0) | |
| Accelerometer-based SB levels, mean (SD); median (IQR) | | |
| Sitting time (hours/day) | 9.6 (1.9); 9.3 (2.8) | |
| Alpha[2] | 1.3 (0.0); 1.3 (0.0) | |
| Usual bout duration (min) | 44.4 (21.2); 40.1 (21.0) | |
| Physical functioning[3], mean (SD); median (IQR) | 8.4 (2.5); 9.0 (2.0) | |
| Cognitive functioning, mean (SD); median (IQR) | | |
| Reaction time (RTI) | | |
| Median duration of the reaction time (ms; negative[*]) | 443.1 (84.1); 428.0 (88.0) | |
| Paired associates learning (PAL) | | |
| Number of correct responses at first attempt | 6.1 (3.4); 5.0 (4.8) | |
| Number of errors (negative[*]) | 42.2 (15.5); 45.0 (27.0) | |
| Spatial working memory (SWM) | | |
| Number of errors (negative[*]) | 21.8 (7.0); 22.0 (8.0) | |
| Strategy use (number, negative[*]) | 9.4 (1.9); 10.0 (2.0) | |
| Delayed matching to sampling (DMS) | | |
| Latency of responses to correct trials (ms; negative[*]) | 4,666.3 (1,852.4); 4,326.0 (2,026.4) | |
| Percentage of correct trials | 66.6 (14.9); 67.0 (20.0) | |
| Rapid visual information processing (RVP) | | |
| Target sensitivity | 0.8 (0.1); 0.8 (0.1) | |
| Overall cognitive functioning[4] | 0.0 (0.6); 0.0 (0.8) | |
| Mental health-related QoL[5], mean (SD); median (IQR) | 73.3 (15.4); 77.5 (16.9) | |

**Table 1** (*continued*)

| Demographics | Total sample (N = 90) | Missing (N) |
|---|---|---|
| Social isolation[6], mean (SD); median (IQR) (negative*) | 42.0 (7.8); 40.4 (10.9) | |

**Notes.**

SD, Standard Deviation; BMI, Body Mass Index (kg/m$^2$); range 18.5 to 24.9, healthy; 25.0 to 29.9, overweight; ≥30, obese; SB, Sedentary Behavior; IQR, Interquartile range; QoL, Quality of Life.

[1] Comorbidity score calculated with NSHAP Comorbidity Index, range 0 to 21.

[2] alpha = cumulative distribution of bout lengths (unitless).

[3] Short Physical Performance Battery for physical functioning score.

[4] Average score of all composite scores, calculated from the average Z-scores of each cognitive test.

[5] RAND-36 Mental Health-Related QoL score.

[6] PROMIS Social Isolation score.

*negative = low score represents better performance/functioning.

**Table 2  Model fit parameters for different profile solutions.**

| Solution | Fit indices | | | | | |
|---|---|---|---|---|---|---|
| | AIC[2] | BIC[2] | SABIC[2] | Entropy | BLRT p-value | Probability range* |
| 2-profile | 2705.856 | 2745.853 | 2695.355 | 0.890 | **0.010** | 0.91–0.98 |
| 3-profile | **2668.792** | **2723.788** | 2654.354 | **0.953** | **0.010** | **0.94–0.99** |
| 4-profile | 2670.098 | 2740.093 | **2651.723** | 0.905 | 0.297 | 0.80–0.97 |

**Notes.**

AIC, Akaike Information Criterion; BIC, Bayesian Information Criterium; SABIC, sample-size adjusted BIC; BLRT, Bootstrap Likelihood Ratio Test.

[2] Lower scores represent better fit.

*Probability range of profile membership (min–max).

Values in bold represent the profile with the best fit for that particular 'fit index'.

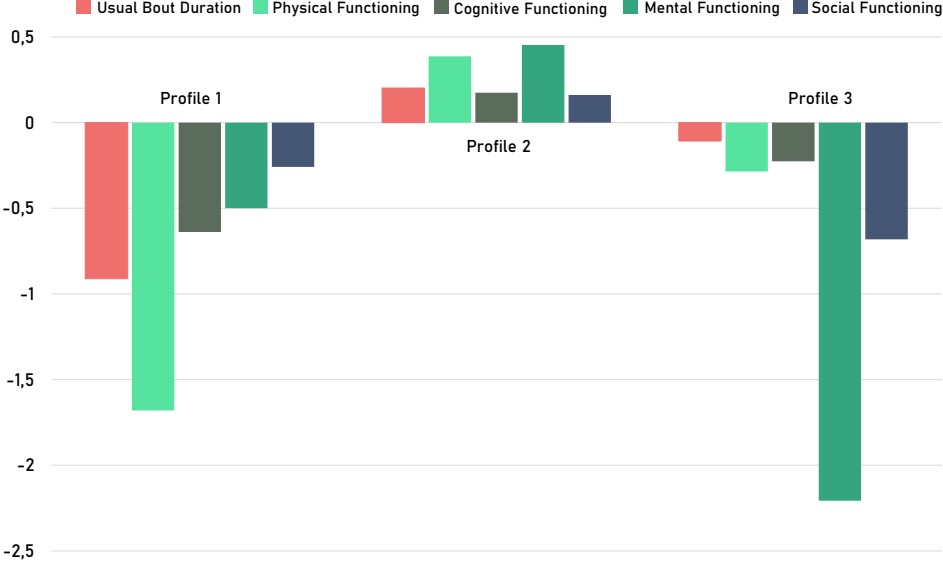

**Figure 1  Latent person-centered profiles based on sedentary pattern and health outcomes (z-scores) in the oldest old.**

**Table 3 Descriptive statistics by profile.**

| | Profile 1 (N = 11) Cognitively and physically frail | Profile 2 (N = 68) Healthy | Profile 3 (N = 11) Lower mental health-related QoL | Overall significance of difference | Between profile (P) differences[$] |
|---|---|---|---|---|---|
| Usual Bout Duration (min), median (IQR) | 47.6 (30.6) | 37.2 (21.1) | 40.2 (13.6) | Kruskal–Wallis = 6.3; $p = 0.04$ | P1 > P2[*] |
| Physical Functioning, median (IQR) | 3.0 (1.5) | 9.0 (2.0) | 8.0 (2.0) | Kruskal–Wallis = 32.2; $p < 0.001$ | P1 < P2[***] P1 < P3[**] P2 > P3[**] |
| Cognitive Functioning, mean (SD) | −0.6 (0.4) | 0.2 (0.5) | −0.2 (0.5) | $F = 11.7$; $p < 0.001$ | P1 < P2[***] |
| Mental Health-Related QoL, mean (SD) | 64.1 (6.0) | 80.2 (7.2) | 40.1 (9.5) | $F = 150.2$; $p < 0.001$ | P1 < P2[***] P1 > P3[***] P2 > P3[***] |
| Feelings of Social Isolation, median (IQR) | 45.7 (10.9) | 40.4 (8.5) | 45.7 (9.0) | Kruskal–Wallis = 5.8; $p = 0.05$ | P2 < P3[*] |
| Age, mean (SD) | 87.1 (3.9) | 84.6 (4.0) | 84.4 (3.8) | Kruskal–Wallis = 3.6; $p = 0.16$ | |
| Female, N (% within profile) | 6 (54.5) | 33 (48.5) | 9 (81.8) | $\chi^2 = 4.26$; $p = 0.12$ | |
| Tertiary education (college/university), N (% within profile) | 2 (18.2) | 22 (32.4) | 3 (27.2) | Fisher's Exact test: $p = 0.74$ | |
| Living in nursing homes, N (% within profile) | 3 (27.2) | 3 (0.04) | 2 (18.2) | Fisher's Exact test: $p = 0.03$ | |
| Single, N (% within profile) | 8 (72.7) | 35 (51.5) | 6 (54.5) | $\chi^2 = 2.9$; $p = 0.24$ | |

**Notes.**

Post-hoc tests for one-way ANOVA: Tuckey's HSD, for Kruskal–Wallis: Dunn's test.

[$]Post-hoc tests (Tuckey's HSD and Dunn's test) were performed to identify significant differences between the three profiles (P1, P2 and P3).

[*]$p < 0.05$.

[**]$p < 0.01$.

[***]$p < 0.0001$.

F, one-way ANOVA statistic.

SB to profile two but had lower physical functioning and mental health-related QoL. A detailed overview of the descriptive statistics per profile can be found in Table 3.

### Sedentary behavior

Median UBD differed between profiles one and two, where profile one had a statistically significant longer UBD, with 47.6 (IQR = 30.6) minutes compared to profile two, with a median duration of 37.2 (IQR = 21.1) minutes ($p = 0.02$). No statistically significant differences were found with profile three.

### Physical functioning

The median physical functioning scores differed significantly among all three profile groups. Profile one had a median score of 3.0 (IQR = 1.5), which was lower than both profiles two ($p < 0.001$) and three ($p = 0.01$). Profiles two and three, with median scores of 9.0 (IQR = 2.0) and 8.0 (IQR = 2.0) respectively ($p = 0.01$), demonstrated higher physical functioning than profile one, with profile two having a higher score than profile three.

### Cognitive functioning

The overall cognitive functioning scores were significantly lower in profile one than in profile two, with mean (SD) scores of −0.6 (0.4) and 0.2 (0.5) respectively ($p < 0.001$). No significant differences were identified with profile three.

### Mental health-related quality of life

The first profile had a mean (SD) score of 64.1 (6.0), which was significantly lower than profile two with a mean (SD) mental health-related QoL of 80.2 (7.2) ($p < 0.001$) and higher than profile three with 40.1 (9.5) ($p < 0.001$). Mental health-related QoL differed also significantly between profile two and three.

## Feelings of social isolation

Differences were found for feelings of social isolation between profile two and three, with median scores of 40.4 (IQR = 8.5) and 45.7 (IQR = 9.0) respectively ($p = 0.05$). No significant differences were found with profile one.

## Differences in socio-demographics across profiles

The Fisher's Exact test showed significant differences between profiles for living in nursing homes ($p = 0.03$). Within profile one, 27% ($N = 3$) of the subgroup lived in a nursing home. For profiles two and three, this was 4% ($N = 3$) and 18% ($N = 2$) respectively.

No significant differences were found for the other socio-demographic variables (age, sex, educational degree and family situation).

## DISCUSSION

This study examined person-centered profiles in the oldest-old (aged 80 years or more) using latent profile analysis (LPA), based on sedentary accumulation patterns and health outcomes, including physical and cognitive functioning, mental health-related QoL and feelings of social isolation. Considering the principles of parsimony, the interpretability of profiles, and the evaluation of multiple fit indices (*Bauer, 2022*), the model with three distinct profiles was considered as the optimal solution. The three distinct profiles were characterized by (1) low cognitive and physical functioning, (2) generally better functioning, and (3) low mental health related QoL. Additionally, the study aimed to explore possible profile prediction based on socio-demographic variables. No meaningful associations were observed for socio-demographic variables, including age, sex, educational degree, and family situation.

The 'cognitively and physically frail' profile was characterized by lower scores on both physical and cognitive functioning. The simultaneous occurrence of cognitive and physical impairment is to be expected, as earlier studies have demonstrated a common tendency of decline in both these domains among older individuals (*Bruce-Keller et al., 2012*). Participants in the 'cognitively and physically frail' profile also exhibited a significantly less favorable sedentary accumulation pattern compared to those of the 'healthy' profile. It is concerning to observe that the median UBD in this profile was 48 min, as prolonged sedentary bouts are associated with reduced blood flow, diminished shear stress, and disrupted postprandial metabolism (*Dempsey et al., 2018*; *Dempsey & Thyfault, 2018*;

*Peddie et al., 2021*). Addressing this behavior in this cognitively and physically frail group of the oldest-old might have the potential to improve overall functioning and mitigate the negative consequences associated with excessive SB.

The 'healthy profile', which was the most prevalent among the oldest-old, scored rather well on all health outcomes, and had the lowest usual bout duration (UBD). Although the profile is labelled 'healthy', it should be noted that this is based on relative values and that older adults in this profile still spent a significant portion of their sedentary time (ST) in prolonged bouts, as evidenced by the median UBD of 37 min. This finding aligns with previous research of Dohrn et al., who found a UBD of 32 min in those aged 80 years and over (*Dohrn et al., 2020*). Thus, despite the relatively high scores on cognitive functioning, physical functioning, mental health-related QoL and low scores on social isolation, the high UBD emphasizes the need for increased efforts to reduce prolonged SB in the oldest-olds (*Copeland et al., 2017*).

Finally, the 'lower mental health related QoL' profile was characterized by low mental health-related QoL and higher feelings of social isolation. The clustering of these concepts aligns with the findings form population-based cohort studies conducted by *Boehlen et al. (2022)* and *Tan et al. (2020)* indicating that lonely older adults often experience a lower mental health-related QoL. Surprisingly, older people belonging to this profile did not exhibit a more unhealthy sedentary accumulation pattern than those in the 'healthy' profile. It should be noted that the association between mental health and sedentary accumulation patterns has not yet been investigated in the oldest-old, and thus, the current results cannot be directly compared with earlier findings. However, a UK study conducted among a general cohort of older adults (mean age 78.4 years) did show that depressed older adults spent more time in longer sedentary bouts compared to their non-depressed counterparts (*Jefferis et al., 2015*).

In an attempt to identify at-risk groups based on SB patterns and health outcomes in the oldest-old, we also examined how specific socio-demographic variables (*i.e.,* age, sex, educational degree, living situation and family situation) related to these three profiles. In general, almost no significant associations were found, meaning that no specific socio-demographic values could predict profile allocation. The difference found between profiles for living in nursing homes carefully suggested that older adults living in nursing homes were mainly present in the 'cognitively and physically frail' profile, which is in line with another study that suggested greater levels of cognitive impairment are associated with increasingly higher odds for moderate or severe physical frailty in older adults residing in US nursing homes (*Yuan et al., 2021*). However, these results are preliminary and should be interpreted with caution as the number of individuals in profile one was low ($n = 11$ of which only three were living in a nursing home). The fact that no socio-demographic variables could be associated with a specific profile, might be partly due to the small sample size, particularly in profiles one and three. Nonetheless, our findings imply that when designing interventions to combat prolonged SB, a more individualized approach based on health outcomes relevant to older adults and the duration of sedentary bouts would be more beneficial than targeting specific socio-demographic subgroups within the oldest-old

population. This approach ensures a focus on the individuals who are most likely to benefit from the intervention, regardless of their demographic characteristics.

This study has some important strengths. First, the study focused on a unique target population (older adults aged 80+ years), which is currently underrepresented in scientific research and no previous studies examined person-centered profiles in this population to the best of our knowledge. As life expectancy has been increasing steadily over the last decades, and our ageing society is imposed with major challenges, it is of utmost importance to examine how we should target our interventions and prevention strategies to keep the oldest-old as healthy as possible for as long as possible. Second, ST, cognitive functioning and physical functioning were measured with valid devices and tests, and mental health-related QoL and social isolation were assessed using valid self-report scales. Next to these strengths, some limitations should be acknowledged. A first limitation is the convenience sampling to recruit the participants, as this limits the generalizability of the results to 'all' older adults aged 80+ years. Nonetheless, the sample showed a good distribution of sex, age, educational attainment, marital status and BMI. However, only a limited number of participants lived in a nursing home, making it impossible to draw firm conclusions. Finally, the total sample size was rather small, which was reflected in the small numbers of participants included in profiles one and three. These limitations emphasize the need for further research with larger and representative samples to enhance the generalizability of the findings across diverse populations of the oldest-old.

## CONCLUSION

In conclusion, this study identified three distinct profiles within the oldest-old population: (1) characterized by low cognitive and physical functioning, (2) exhibiting generally better functioning, and (3) marked by low mental health related QoL. These classifications were based on sedentary accumulation patterns and geriatric-relevant health outcomes. The findings highlight the association between lower levels of physical and cognitive functioning and unhealthy sedentary patterns, with half of their sedentary bouts lasting at least 48 min.

Our study found no clear associations with specific socio-demographic factors. Instead, our results suggest that interventions targeting prolonged SB, should adopt a more individualized approach based on health outcomes and the duration of sedentary bouts rather than focusing on specific socio-demographic subgroups within the oldest-old population.

Further investigation is imperative to enhance the understanding of these subgroups, and studies with larger sample sizes are essential to potentially identify associations between socio-demographic factors and these at-risk groups. This will contribute to a nuanced comprehension of the factors influencing SB and geriatric health-related outcomes in the oldest-old population.

## ACKNOWLEDGEMENTS

The authors would like to acknowledge all researchers and students involved in the data collection, and all participants.

### Funding

Elien Lebuf, Sofie Compernolle and Jelle Van Cauwenberg are supported by Research Foundation Flanders (project numbers G020620N, 12Z4221N and 12I1117N). The funders had no role in study design, data collection and analysis, decision to publish, or preparation of the manuscript.

### Grant Disclosures

The following grant information was disclosed by the authors:
Research Foundation Flanders: G020620N, 12Z4221N, 12I1117N.

### Competing Interests

The authors declare there are no competing interests.

### Author Contributions

- Elien Lebuf conceived and designed the experiments, performed the experiments, analyzed the data, prepared figures and/or tables, authored or reviewed drafts of the article, and approved the final draft.
- Delfien Van Dyck conceived and designed the experiments, authored or reviewed drafts of the article, and approved the final draft.
- Laura Van de Velde performed the experiments, authored or reviewed drafts of the article, and approved the final draft.
- Melanie Beeckman conceived and designed the experiments, authored or reviewed drafts of the article, and approved the final draft.
- Jelle Van Cauwenberg conceived and designed the experiments, analyzed the data, authored or reviewed drafts of the article, and approved the final draft.
- Sofie Compernolle conceived and designed the experiments, performed the experiments, authored or reviewed drafts of the article, and approved the final draft.

### Human Ethics

The following information was supplied relating to ethical approvals (i.e., approving body and any reference numbers):

The study was approved by the ethical committee of the Ghent University Hospital (BC-08641).

### Data Availability

The data and scripts are available at Open Science Framework: Lebuf, Elien, Sofie Compernolle, and Delfien Van Dyck. 2023. "Publication." OSF. November 7. https://osf.io/g2cn7/.

### Supplemental Information

Supplemental information for this article can be found online at http://dx.doi.org/10.7717/peerj.17505#supplemental-information.

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
