# Peer review of "Sedentary patterns and health outcomes in the oldest-old: a latent profile analysis"

_PeerJ, doi:10.7717/peerj.17505_

## Round 0.1 · original submission · Major Revisions

As you can see, we have had comments from two very helpful reviewers, both of whom have requested changes to your manuscript. Each of their comments warrant careful attention. Each comment should either be addressed through tracked revisions to your work or through an explanation of why you feel no changes are warranted for that particular comment(s).

Reviewer #1’s comments on missing data and the discussion are particularly important, but all of their comments should help you to improve your work. Reviewer #2 similarly makes important comments about the discussion, particularly their comments around nursing home residents, and again all of their comments should be helpful to you. Both reviewers have been very helpful in suggesting changes to improve the readability of your manuscript.

I’d like to see a brief explanation of the recruitment process in the abstract’s methods as these are key to understanding the generalizability of the results.

The use of a number of model fit indices (Lines 205–210) leads to a question of which would have been used where there were conflicts (this isn’t unusual for BIC and AIC, for example). Table 2 is clear on balance, but if your a prior criterion had been SABIC over all others, you might have gone with the 4 class model. Also, with LPA it is common for there to be a minimum number or proportion of observations per class and for subject-matter considerations to be used in selecting the number of classes. Were either of these methods used here? I think readers will want some references and/or argument for the small numbers in some classes.

Please explain in the methods why/when you used one-way ANOVAs or Kruskal-Wallis (Lines 212–214) and what post-hoc tests were used. Were overall (e.g. Wald) tests also used? These appear in Table 3 (I’m not convinced that the test statistics here are going to help readers though) but the pairwise comparisons are then not clear here. Perhaps you could use the superscript letter system (where classes with the same letter for that characteristic are not statistically significantly different). Similarly, please explain in the methods the rule used to switch to Fisher’s Exact Test (Lines 216–217).

Given that you have 90 participants, I suggest that integer percentages (c.f. Line 227, etc.) be used since each participant contributes more than a full percent to these values.

While you identify when you used medians (e.g., Lines 252–253), the values with the plus-minus sign are not clear. Readers could wonder if these are IQRs, SDs (which wouldn’t match well with medians), or something else. Table 1 is clear in this respect.

I strongly urge against the use of “borderline significant” (Lines 261 and 269). This has been discussed in the (bio)statistical literature as difficult to interpret (which side of the border?) and as encouraging the dichotomization into statistically significant versus not.

Figure 1’s x-axis is not ordinal so I don’t think the line graph approach is appropriate here.

I look forward to seeing a revised version of your very interesting manuscript in due course.

Reviewer 1 ·

Basic reporting

This manuscript reports on a cross-sectional study designed to identify profiles of oldest-old adults based on their sedentary behavior patterns and health outcomes and to examine the association between profiles and sociodemographic variables. The manuscript is easy to read, and the English language is appropriate except for a few minor issues (see minor editorial notes below).

This research addresses an important clinical issue that has the potential to affect a larger and larger number of older adults as the population ages. The study of sedentary behavior is a relatively new line of research, and few studies have focused on the oldest-old adults. The introduction provides a solid rationale to justify the need for this research and the specific methods employed.

Minor editorial issues
1. Line 47-49. The definition of “accumulation patterns of sedentary time” is quoted from a credible source but is awkward and vague. A better definition or description of this term is needed.
2. Line 51. It is not necessary to define a “sedentary interruption or break.” These terms do not require a definition, and the definition breaks up the flow of the paragraph.
3. Line 57. Replace “with” with “by.”
4. Lines 70-73. The importance of tailored interventions is not highly relevant to this research and tends to interrupt the flow of this paragraph. Consider deleting it.
5. Line 95. The meaning of “novice care situation” is not clear. Consider using a term that is widely understood.
6. Line 108. Insert “the” in front of gold standard.
7. Line 174. One or two words are missing here. “Scoring was done conform the PROMIS…”
8. Lines 233-238. Abbreviations used in Table 1 should be included in the table and do not need to be repeated in the body of the manuscript.
9. Line 295. The phrase “next to the diminished physical and cognitive function” serves no purpose. Consider deleting it.

Experimental design

This is original research. with clear aims that are systematically addressed in this report.

The methods used to measure sedentary behavior and health outcomes are appropriate. Sedentary behavior was measured with the activPAL device, which is considered to be the gold standard measure of sedentary behavior. Analyzing these data to produce meaningful results can be challenging, but the combined use of total sedentary time and usual bout duration provides an informative description of participants’ behavior. Fourteen of 112 participants provided incomplete activPAL data; this is unusual for the activPAL device, and the reasons for missing data are not described.

The health outcome variables provide a comprehensive assessment of functioning in multiple relevant domains. Well-established, reliable, and valid measures were used to measure each variable, and each measure was clearly described and documented, thereby facilitating the interpretation of results.

The sample size is relatively small, and the authors acknowledge this, but so little research has been done with this population that it is reasonable and makes a useful contribution that advances the field. Participants provided written consent prior to engaging in the research.

The data analysis is appropriate and described in exquisite detail in 1 1/2 pages of text. Consider condensing the description without losing the most important elements of the analytic process. The tables and figure are clear and contribute to the quality of the manuscript.

Validity of the findings

The primary results are valid, but profiles 1 and 3 consist of very small groups, 11 subjects each, and the interpretation of results from these groups must be considered preliminary. This issue is addressed below in the review of the discussion and conclusions section.

In general, the discussion addresses the most important findings and places them within the context of prior research, but there are several limitations, and further refinement is needed. The introductory paragraph is diffuse and does not mention the main findings. It focuses too much on the methods and on the importance of tailoring interventions without describing the most important findings. More broadly, the discussion focuses too much on the importance of tailoring interventions. This research was not about tailored interventions, and the discussion would be strengthened by deleting most, if not all, of the discussion about the importance of tailoring interventions. It goes beyond this research. Other suggestions for interventions also go beyond the research. Specifically, lines 302-306 introduce the potential benefits of an intervention interrupting sedentary time, and lines 346-351 discuss the importance of individualized interventions. Profiles 1 and 3 had only 11 subjects each; with these small numbers, associations with health outcome variables could be unreliable. Furthermore, examining subgroups within either of these profile groups is inappropriate, including the discussion and conclusions about the 3 of 11 nursing home residents in profile group 1. Please delete all content related to the subgroup of 3 people in profile group 1 who reside in a nursing home from the discussion and the conclusion.

The conclusion does not adequately address the most important findings from this research: a) three profiles were identified and the nature of these profiles, and b) the lack of an association with sociodemographic variables. Please revise the conclusion to focus on these issues without going beyond the research.

Additional comments

None.

Reviewer 2 ·

Basic reporting

Overall very good, however some wording/grammar issues throughout - see detail in additional comments section.

Experimental design

Research question is well defined and useful in the topic area of sedentary behaviour and older adulthood. Methods are clear and well-described. I have raised question over not collecting data on bereavement and ethnicity within the demographics.

Validity of the findings

Mainly good and supporting data is provided. However, I do question to significance of the findings relating to nursing home participants, as explained in further detail in the attached document.

Additional comments

The manuscript covers an important research area and adds novel value to the literature. It is mostly well-written and methods seem sound, however as a mainly qualitative researcher I do not feel qualified to comment upon the soundness of the statistical analysis presented. My comments are mostly minor grammatical and wording issues, however there are a couple of more major points that I think need addressing.

Abstract
• Background:
o First line - Aged 80 ‘years’ and above (or aged 80+ years). This could be amended when stating age throughout manuscript for added clarity.
o SB in the second line should be written out in full (as not abbreviated elsewhere in abstract)
o Need to explicitly specify that the ‘oldest old’ is referring to those aged 80 years and above
o Aims – be clear here that you are talking about the oldest old age group not older adults generally.
o Would be good to specify you are focusing on the oldest-old somewhere in background rationale (though understand you are under word limit pressure)
• Methods: Very clear – perhaps specify ‘Associations of these profiles with socio-demographics were…’
• Results and conclusions clear.

Introduction
• Line 45 – does the data for older adults aged 60 years and above include that of 80 and above? Or is it 60-80?
• Could make a little more explicit the difference in definition of sedentary behaviour (SB) and sedentary time (ST) – unclear if these are used interchangeable or not?
• Line 57: decreased by 7% (rather than with)
• Line 58: change ‘men’ to ‘those’
• Line 60: insert ‘have’ before ‘mainly’
• Line 61: the general population of older adults being over 65 years – delete ‘being’ here. Would also be good to mention that this is in line with international recommendations on physical activity/sedentary behaviour from the World Health Organization, who use 65 year to signify ‘older adult’ age category.
• Line 64: Check citation format is correct
• Line 69: Be more explicit what you mean by saying a ‘multi-disciplinary’ approach – examples of disciplines you are referring to here?
• Line 70: ‘By taking this perspective…’ – be explicit what ‘this’ perspective is, do you mean holistic/multi-disciplinary?
• Line 74/75: Check citation formatting
• Line 72/73: ‘previous studies have indicated that tailored interventions tend to yield greater success compared to non-tailored ones’ – this needs a reference. Further, it would be good to give some insight into previous sedentary behaviour change interventions in older adults here
• Line 74: Not sure ‘underscored’ is the right word here – underestimated? Can you provide some evidence/stats for this?
• Line 74/75: check citation formatting
• Line 75/76: ‘determining the link between socio-demographics and these at-risk profiles can provide valuable insights’ – be clear that when your talking about profiles you mean SB patterns and health outcomes – often by ‘profiles’ we are referring to socio-demographics, so this becomes confusing unless spelled out explicitly.
• Line 77: This sentence is a little vague – consider revising for clarity
• Line 81/82: Either ‘these being’ or delete ‘being’ and put ‘physical, cognitive and psychosocial functioning’ in parentheses.
I think there are a couple of areas of literature where you could include more detail and evidence, and some points of clarification, but overall this you provide and clear and logical rationale for your study here.

Methods
• Line 95: Could you explain what is meant by ‘novice care situation’?
• Line 115: ‘to exclude sleep from the analysis’
• Line 125: check citation format
• Line 179: Missing ‘The’ at beginning of sentence (before ‘following)
• Line 188: Analysis should be plural
• Line 198: need ‘these’ before ‘being’
• Line 217: need ‘The’ before ‘following’

Methods are thorough and clear (although I cannot comment on the statistical approach here). I did wonder if you had considered asking about bereavement here? This links with sedentary behaviour and health status so would have thought it would be looked at in demographic – maybe something to reflect upon? I was also surprised that ethnicity was not mentioned in socio-demographics, was there a reason for this?

Results
• Line 230: change ‘was’ to ‘were’
• Line 267: Can you state explicitly the direction of difference between profiles two and three here.
• Line 276: need ‘these’ before ‘being’
• See below point in discussion re nursing homes – significance needs to be made explicit.
Clearly presented results section and interested findings with the profiles. I wonder if it might be worth having a short descriptive overview of the three profiles so the reader can get a qualitative sense of each distinct grouping?

Discussion
• It would be good to summarise/give a clear outline of all findings in the opening of your discussion
• Line 294: “deterioration in these domains frequently goes hand in hand in older individuals” – I find this phrasing a little odd, consider revising
• Line 304: check citation format
• Line 314: check citation format
• Line 314: replace ‘underscores’. Also check citation format.
• Line 328: I’m finding it difficult to give weight to the findings relating to living in a nursing home. As profile two was ‘healthy’ then you’d very few of these were living in nursing home. Was the difference between profiles one and two/three a significant difference? Or was the significance between 2 and then one/three? And was there a significant difference between one and three at all? (This needs clarifying in your results). With only N of 11 and this being spit 3 to 2 between the two profiles, I’m not sure this can be viewed with much weight. It seems an over-exaggeration to say they were mainly present in the cognitively and physically frail profile. I see this is acknowledged as a limitation later in the discussion, but I suggest you being a little more tentative in your stating of findings here.

---

## Round 0.2 · Minor Revisions

Thank you for your responses and revisions. Reviewer #1 has provide a small number of comments, most of which aim to improve readability. Reviewer #2 has raised no new points.

Note that the paired samples Wilcoxon is not the correct post-hoc test for Kruskal-Wallis. Please use Dunn’s test or provide a justification (with a reference) to another test if you prefer to use a non-standard test that readers might not be familiar with. While the article is written in the context of using Stata, the Introduction section from https://doi.org/10.1177/1536867X1501500117 is a useful reference here.

Reviewer 1 ·

Basic reporting

This is the review of a revised manuscript.

This manuscript is much improved, and previous reviews were adequately addressed. A few minor issues remain with respect to clarity and/or English language usage. They are listed below. Note line numbers refer to the tracked version “Show All Markups.”

Line 75. Please change this sentence's second “which” to “and.” With this change, it will read “years or older, which includes those aged 80 and above (Webster et a. 2021), and underlines the”
Lines 77-84 of the track version of the manuscript. This text is not clear. I recommend working with a fluent English-speaking editor to clarify this section.
Line 256. The sentence that starts at the end of this line is not clear. “Scoring was done conform the following PROMIS scoring system:” Something like this would be clearer. “Scoring was done according to PROMIS guidelines:”
Line 508 - 509. The first sentence in this paragraph is redundant to the prior paragraph and unnecessary.
Line 682. In the conclusion, it was noted that “a significant portion of their sedentary bouts lasted at least one hour.” The mean UBD was actually 44.4±21.2 minutes, and this was the first mention of one-hour bouts. New information should not be introduced in the conclusion.

Experimental design

No changes were required to this section.

Validity of the findings

No changes required to this section.

Additional comments

N/A

Reviewer 2 ·

Basic reporting

No further comments

Experimental design

No further comments

Validity of the findings

No further comments

Additional comments

The authors have made all necessary amendments to the manuscript and I am happy this is now suitable for publication. I feel this study will make an important and novel contribution to a research area where current literature is sparse.

---

## Round 0.3 · Minor Revisions

Thank you for your revised manuscript and your responses to the reviewers’ and my comments.

I have made some final editing comments below. If there were fewer, I’d accept the manuscript now and suggest that you address these in proofing, but I think there are enough comments to make that something we need to finalise now. I would expect to be able to quickly accept your manuscript once these are addressed.

1. You use “the oldest-old” in the title, but not always elsewhere (e.g., Lines 19, 21, etc.) although sometimes the hyphen does appear (e.g., Lines 35, 71, etc.). Is there a pattern I’m missing or ought these to be consistent?

2. You could add “Kruskal-Wallis” to the methods in the abstract (Line 29), or you could remove “using one-way ANOVAs and chi2-tests” from there as I suspect most readers of the abstract will be less interested in these details, which are well described later.

3. On Line 91, you say “possible predictions” where I think “possible predictors” would be easier to read.

4. On Line 92, you say “profile allocating” where I think “profile membership” would be easier to read. Related to this, you could delete “allocation” from Line 299.

5. On Line 104, you say “informative”. Do you mean “information” or something else?

6. On Line 188, you have a link. Could you rewrite this so the URL is included in a footnote? That way, it’s still useful for printed versions.

7. On Line 191, you say “of which” where I think “from which” would be clearer.

8. On Line 228, you describe living situation as “dichotomous”, but the other categorical variables here are also dichotomous so it seems that the label could be removed (or added to all other binary/dichotomous variables here).

9. On Line 237, could you add “(±SD)” into here, i.e., “mean (±SD) age of 84.9±4.0” for clarity. See also Lines 240 and 241. You do this on Lines 276, 279, etc. and in Tables 1 and 3 already. This will also mirror your descriptions of IQRs, e.g., Line 266 and in the tables.

10. For the percentages here (Lines 238–240) and elsewhere, including Table 1, I think integer percentages would be sufficient (as you do elsewhere, e.g., Lines 250–251 and following). With n=90, each person contributes 1.1%, so the values after the decimal point are not really needed. Also, there seemed a missing “%” after “55.1” (which I think should become “55%”).

11. For Lines 287–289, you describe a Chi-squared test, but based on the rule on Lines 225–226, I was expecting Fisher’s Exact test instead. The alternative rule often seen that at least 80% of cells should have expected counts of 5 or more would also trigger Fisher’s Exact test here.

12. On Line 297, you could drop “most” from “most optimal solution” as optimal already has that singular aspect built in (like “the highest place”).

13. This is just stylistic, but “rather well” seems more natural to me than “rather good” on Line 312.

14. For Table 1, perhaps move the leading “%” signs and make them “(%)” after the variable names?

---

## Round 0.4 · accepted · Accept

Thank you for your revised manuscript and responses. I'm delighted to accept your work and look forward to it generating discussion about this important age group.

My only suggestion would be that the presentation of percentages as integers didn't get applied to Table 1, but I will leave this for you to act on as you see appropriate.

Congratulations on your success!